# Effects of Repetitive Transcranial Magnetic Stimulation Applied over the Primary Motor Cortex on the Offset Analgesia Phenomenon

**DOI:** 10.3390/life15020182

**Published:** 2025-01-26

**Authors:** Elisa Antoniazzi, Camilla Cavigioli, Vanessa Tang, Clara Zoccola, Massimiliano Todisco, Cristina Tassorelli, Giuseppe Cosentino

**Affiliations:** 1Translational Neurophysiology Research Section, IRCCS Mondino Foundation, 27100 Pavia, Italy; elisa.antoniazzi@mondino.it (E.A.); camilla.cavigioli01@universitadipavia.it (C.C.); vanessa.tang01@universitadipavia.it (V.T.); clara.zoccola01@universitadipavia.it (C.Z.); massimiliano.todisco@mondino.it (M.T.); 2Department of Brain and Behavioral Sciences, University of Pavia, 27100 Pavia, Italy; 3Headache Science and Neurorehabilitation Center, IRCCS Mondino Foundation, 27100 Pavia, Italy

**Keywords:** repetitive transcranial magnetic stimulation (rTMS), offset analgesia (OA), primary motor cortex (M1), endogenous pain modulation, quantitative sensory testing (QST), somatotopic organization, chronic pain management

## Abstract

In this study, we investigate the effects of high-frequency repetitive transcranial magnetic stimulation (rTMS) applied over the left upper limb primary motor cortex (M1) on the offset analgesia (OA) phenomenon, a measure of endogenous pain modulation. In particular, we aim to determine whether rTMS influences OA differently in the forearm region, corresponding to the stimulated cortical area, compared to the trigeminal region. Twenty-two healthy volunteers underwent three experimental sessions: a baseline session without stimulation, an active rTMS session, and a sham rTMS session. Quantitative sensory testing (QST) paradigms, including warm and cold detection thresholds, heat pain threshold corresponding to a visual analogue scale (VAS) score of approximately 50–60 out of 100 (Pain_50–60_), and constant and offset trials, were assessed in both the forearm and trigeminal regions. The results revealed that active rTMS significantly enhanced the OA phenomenon in the forearm during the late phase, while no significant effects were observed in the trigeminal region. These findings suggest that rTMS may modulate central pain mechanisms in a body region-specific manner, potentially linked to the somatotopic organization of M1. This study points to possible mechanisms of action of rTMS for pain relief, highlighting the importance of region-specific effects in chronic pain treatment. Further research is needed to investigate the underlying mechanisms and clinical applicability of rTMS in patients with chronic pain conditions, especially when OA is compromised.

## 1. Introduction

Pain is a multidimensional phenomenon influenced by affective, psychological, and sociocultural phenomena mediated by neuronal activity across networks in the central (CNS) and peripheral nervous systems (PNS). Endogenous pain modulation, involving numerous neurotransmitters and receptors, plays a critical role in nociception, and its dysfunction can drive the transition from acute to chronic pain [1,2]. Pain is further influenced by cognitive and motivational factors, as well as psychological states like anxiety and depression, which can exacerbate pain through mechanisms such as central sensitization and altered activity in pain-related neural circuits [3,4,5].

The efficiency of pain modulatory mechanisms can be assessed using specific quantitative sensory testing (QST) paradigms, including the conditioned pain modulation (CPM), which reflects the ability of one noxious stimulus to suppress the perception of a subsequent one, and the offset analgesia (OA) phenomena [1,2,6,7]. OA is defined as a significant reduction in pain perception following a minimal decrease in the intensity of a heat painful stimulus [6]. It is considered an expression of the activation of temporal sharpening mechanisms of dynamic nociceptive stimuli [7]. These phenomena are crucial for understanding the differential modulation of pain in various conditions, such as neuropathic and inflammatory pain, where dysfunctions in CPM or OA mechanisms may contribute to the persistence of pain [7]. Moreover, evaluating pain modulatory mechanisms has proven particularly useful in distinguishing chronic pain phenotypes by identifying differences in inhibitory and facilitatory responses across patient populations. For instance, it aids in differentiating central and peripheral pain mechanisms or stratifying patients based on the function of their endogenous pain modulation systems, thereby supporting personalized treatment approaches [8,9,10]. 

Recently, it has emerged that high-frequency repetitive transcranial magnetic stimulation (rTMS) applied to the primary motor cortex (M1) can reduce pain perception in patients with chronic pain [11,12,13,14,15]. However, the mechanisms underlying this pain reduction remain poorly understood. In healthy subjects, increased excitability of the corticospinal motor system induced by high frequency rTMS can lead to more efficient inhibitory modulation of pain [16]. Several hypotheses exist regarding the mechanisms through which rTMS reduces pain, including thalamic activation, which suppresses sensory information transmission via the spinothalamic pathway, as well as the activation of specific brain areas involved in descending pain modulation systems, such as the brainstem, anterior cingulate cortex, and dorsolateral prefrontal cortex (DLPFC) [17,18]. Additionally, reward mechanisms involving the putamen, medial prefrontal cortex, and nucleus accumbens may play a role in suppressing the negative emotional and behavioural aspects associated with neuropathic pain [19,20,21].

In a previous study, we observed that high frequency rTMS applied to M1 can induce a delayed and attenuated OA phenomenon at the thenar eminence, where OA is generally not detectable [16,22]. Considering that it is still unknown to what extent the modulatory effects of rTMS applied to the primary motor area are focal within the corresponding body region or spread to other regions, in this study we evaluated whether the effects of active (real) or sham (placebo) rTMS over the M1 hand representation area are region-specific. Thus, we investigated the OA phenomenon in both the forearm and trigeminal regions of healthy individuals, where OA is easily measurable. We hypothesized that applying rTMS to the M1 area corresponding to the upper limb would enhance OA in the forearm region, with less pronounced effects in the trigeminal region.

## 2. Materials and Methods

### 2.1. Subjects

We enrolled 22 right-handed healthy volunteers (18 females; mean age ± SD: 23.5 ± 1.6 years; age range: 20–27 years), all naïve to rTMS and QST procedures.

All subjects were clinically tested to exclude polyneuropathy through sensory and motor assessments and a detailed medical history. None had chronic pain or neurological disorders, and women’s sessions were kept consistent with their menstrual phase.

Exclusion criteria were the same as in our previous work [16], including pregnancy, breastfeeding, severe systemic or psychiatric disorders, skin issues at the stimulation site, medications affecting pain perception or neuroexcitability (including antiepileptic and antidepressant drugs), poor sleep before testing, and contraindications to rTMS, such as epilepsy and implanted devices.

Due to the exploratory nature of this study, the sample size was not determined through formal calculation but determined based on available resources and the existing literature. Specifically, in our prior work [16], we showed that individual rTMS sessions administered to the human cortex can significantly modulate the offset analgesia phenomenon in a population of subjects with comparable sample size. Similarly, Taylor et al. [23] demonstrated that individual rTMS sessions administered to the human cortex can produce analgesic effects in a population with a relatively small sample size.

### 2.2. Study Procedures

Each subject participated in three distinct experimental sessions conducted on different days, with at least one week between each session (Figure 1). In each session, we began by evaluating the warm detection threshold (WDT), cold detection threshold (CDT), and heat pain threshold corresponding to a visual analogue scale (VAS) score of approximately 50–60 out of 100 (Pain_50–60_). We then applied three constant tests (CTs) and three offset analgesia (OA) tests to the same body region. All measurements were recorded first in one region (forearm or trigeminal area) and then in the other according to a randomized order. The three experimental sessions were conducted in a balanced and randomized order, determined by a computer-generated algorithm. These sessions included a baseline session, which involved only quantitative sensory testing (QST) paradigms (i.e., WDT, CDT, Pain_50–60_, CT, and OA), and two rTMS sessions, in which the same QST measurements were assessed immediately after either sham (placebo) or active (real) rTMS. The baseline session was conducted separately to avoid potential confounding factors: (1) conducting repetitive tests within a short time frame in the same session could influence responses or induce participant fatigue, and (2) consistently performing baseline measurements immediately prior to post-rTMS assessments could introduce bias. By scheduling three separate and randomized sessions, we ensured that the influence of sequence effects was minimized.

### 2.3. Quantitative Sensory Testing Assessment

QST assessments took place in a quiet room maintained at a temperature of 22–24 °C, conducted by two skilled neurophysiology technicians. Prior to the examination, participants were briefed on the visual analogue scale (VAS) assessment method and the experimental evaluations, with demonstrations performed on their left hand.

The skin temperature on the right forearm and forehead of each participant ranged between 34 °C and 37 °C. The experimental procedures employed a 30 × 30 mm air-cooled heat probe, designed to deliver a stable temperature range from 14 to 50 °C with a ramp (0.1–2 °C/s) and hold technique. The probe was connected to a Q-sense Conditioned Pain Modulation device (Medoc, Ramat Yishai, Israel). During all measurements, participants were asked to keep their arm or head still to avoid any movement that could compromise the accuracy of the evaluations.

WDT and CDT were measured using the ‘method of levels’. Subsequently, the Pain_50–60_ was calculated in two different ways: first, by means of the ‘method of limits’, and then by the ‘method of levels’ using a continuous analogue-to-digital converter of VAS (CoVAS, Medoc, Israel), following standardized procedures [24,25,26,27,28].

For WDT and CDT, patients were asked to respond with ‘yes’ or ‘no’ based on whether the hot or cold stimulus was perceived. The interval between stimuli was randomized within a range of 4 to 6 s. The initial temperature change was set at 3 °C. If a participant responded with ‘yes,’ the subsequent stimulus was presented with a step size reduced by half. Conversely, if the response was ‘no,’ the step size was doubled. With every change in response direction, the step size was either halved or doubled accordingly. The process continued until the step size was reduced to 0.1 °C, at which point the threshold was determined by averaging the temperatures corresponding to the final ‘yes’ and ‘no’ responses. A standard series of level-based stimuli ultimately produced a single threshold value. For all trials, the thermode’s baseline adaptation temperature was maintained at 32 °C, with temperature changes occurring at a rate of 1 °C per second.

When calculating the Pain_50–60_ using the method of limits, the mean threshold temperature was determined based on three stimuli, during which the temperature increased at a constant rate of 1 °C per second. Participants were instructed to press a mouse button with their free left hand as soon as they felt the temperature reached a pain intensity corresponding to a VAS score of 50–60 mm as recorded by the CoVAS, where 0 indicated ‘no pain’ and 100 represented ‘the most intense pain imaginable’. The stimuli were immediately stopped when the participant pressed the mouse button. Throughout all trials, interstimulus intervals of 8–10 s were maintained, and no hints were given to the participant at the start of the stimulus. As for the WDT and CDT assessment, the adaptation temperature of the thermode was set to 32 °C, with an increase and decrease time of 1 °C/s. Afterwards, Pain_50–60_ was measured using the method of levels. In this case, subjects were exposed to a starting temperature ranging from 41 °C to 44 °C for 5 s (with an increase and decrease time of 2 °C/s and 1 °C/s, respectively), based on their individual Pain_50–60_ values assessed by the method of limits, and were asked to indicate the pain they perceived using the CoVAS, as previously described. The temperature was then increased or decreased by 0.5 °C until a VAS of 50–60 mm was obtained. Interstimulus intervals of 20 s were maintained, beyond the increase and decrease time. After the Pain_50–60_ assessment, three OA trials and three constant trials were administered using a well-established procedure [6,24,29,30,31,32]. Average values were computed to derive single scores for OA and constant trials. OA trials consisted of a 5 s interval at the individualized Pain_50–60_ assessed by the method of levels (T1), 5 s at a temperature 1 °C higher than T1 (T2), and 20 s at the same temperature as T1 (T3). Constant trials consisted of a 30 s stimulus at Pain_50–60_ determined by using the method of levels.

When evaluating the OA phenomenon, we referred to the Pain_50–60_ values assessed using the method of levels rather than the method of limits because the former method allows for a more precise estimation of the thermal pain threshold and reduces the variability that may be present with the method of limits [33,34]. Although the method of levels is time-consuming, it can be expedited by initially assessing Pain_50–60_ with the method of limits, as we did in the present study, as this provides an approximate value that can guide the more accurate method of levels procedure [34].

For participants with an average Pain_50–60_ exceeding 48 °C, the intensity of stimulation was fixed at 48 °C for both constant and OA trials (T1 and T3), with a minimum interval of 60 s between trials. Temperature increase and decrease rates were set at 2 °C/s and 1 °C/s, respectively. The six trials were conducted following two pseudorandomized sequences (i.e., OA-CT-OA-OA-CT-CT or CT-OA-CT-CT-OA-OA). During each trial, participants were instructed to evaluate the pain intensity using the CoVAS. Interstimulus intervals between trials were at least 30 s, and a new trial was not applied until any sensation of pain in the stimulated area had completely disappeared, with a minimum waiting time of 20 s after the pain had subsided. Participants were not informed about the details of the OA paradigm or the goals of this study. They were instructed to notice even slight pain differences. The thermode was securely attached to the forearm with a Velcro elastic band, positioned to maintain good skin contact without discomfort. They were guided on how to respond correctly, and temperature changes on the computer screen were hidden from their view during testing.

### 2.4. Methods of Stimulation

Magnetic stimulation was applied using a 70 mm figure-eight air-cooled coil connected to a Magstim Rapid2 Stimulator (Magstim Company Ltd., Whitland, UK). Participants were seated comfortably in a reclining chair, adjusted to a slight recline with a headrest to ensure stable head positioning during the rTMS session. They were instructed to stay relaxed throughout the procedure, wearing earplugs and a snug swimming cap marked to identify the stimulation site for precise coil placement. For active stimulation, the coil was oriented posteroanteriorly over the left motor cortical area targeting the contralateral abductor pollicis brevis (APB) muscle [35]. The examiner continuously monitored the participants to maintain proper positioning throughout the session. During sham stimulation, the coil was angled 90° from the scalp, maintaining one wing’s contact at the same location, simulating TMS effects like sound and touch without biological impact [36].

During the experiment, subjects could not view the coil’s positioning. Each rTMS session included 30 trains of TMS pulses at 10 Hz for 10 s (100 pulses per train) with a 20 s pause between trains, reaching a total of 3000 pulses in 15 min (Figure 2) [11]. The stimulation intensity was adjusted to 80% of the resting motor threshold, which was determined as the minimum intensity capable of producing a response of at least 50 µV in the relaxed right APB muscle in 50% or more of ten trials [11]. EMG signals were captured via pre-gelled surface electrodes (Ambu Neuroline 715) spaced 3 cm apart over the muscle. The EMG setup included a 10–1000 Hz bandpass filter and a display gain of 50–200 µV/cm, using a Cadwell Sierra Summit EMG System (Cadwell Industries, Inc., Kennewick, WA, USA). Coil position was monitored continuously to maintain consistent placement, with stimulation following established safety guidelines [37,38].

### 2.5. Statistical Analyses

To determine whether to use parametric or non-parametric statistics or apply a logarithmic transformation, the Kolmogorov–Smirnov test was initially performed on all data. As all variables followed a normal distribution, parametric methods were applied in all cases. A one-way analysis of variance (ANOVA) with a within-subjects factor ‘condition’ (three levels: baseline, active, and sham) was conducted to assess differences in WDT, CDT, and Pain_50–60_ calculated using both the limits and levels methods, on both the forearm and forehead across the three experimental sessions.

To distinguish between adaptation or sensitization effects and offset effects, the OA magnitude was calculated by subtracting the average pain ratings at 1 s intervals during the OA trial from those in the constant trial (∆OA) [24].

Two-way ANOVAs were conducted to evaluate differences in ∆OA values between sessions for the forearm and forehead. The analysis included two within-subject factors: ‘session’ (three levels: baseline, active, sham) and ‘time’ (36 levels, corresponding to VAS measurements taken at 1 s intervals). A 36 s time window was selected to include the after-sensation phase, capturing the rate of VAS value decrease following the trial during the gradual reduction in the intensity of the painful heat stimulus (referred to as T4).

Finally, as additional supplementary analyses, three separate two-way ANOVAs with within-subject factors ‘condition’ (two levels: OA and CT) and ‘time’ (36 levels, corresponding to VAS at 1 s intervals) were performed for each session (baseline, active rTMS, and sham rTMS) for both the forearm and the trigeminal region. These analyses aimed to measure the differences at each second between the VAS values recorded during the CT and OA trials across the three different sessions and to evaluate the temporal characteristics of the OA phenomenon including the after-trial phase.

For all analyses, significance was set at *p* < 0.05. All statistical analyses were performed using StatSoft Statistica version 14.2.0 (StatSoft, Inc., Tulsa, OK, USA).

## 3. Results

All enrolled subjects completed the planned experimental evaluations, and both QST and rTMS were well tolerated without significant side effects during or after each session.

No significant differences were found in WDT, CDT, or Pain_50–60_ (calculated using both the limits and levels methods) across the different experimental sessions (baseline, active, and sham) on both the forearm (Figure 3) and the forehead (Figure 4).

ANOVA conducted for ∆OA values recorded on the forearm (Figure 5) showed a significant interaction between factors (F(70,1470) = 1.7, *p* = 0.0003), prompting a post-hoc analysis using Duncan’s test. This analysis revealed significantly higher values, indicating a greater OA phenomenon, in the real condition compared to both the sham condition (*p* < 0.05 at every second from the 26th to 33rd s, with the highest difference observed at the 31st s, *p* = 0.02, corresponding to ∆OA values of 13.1 and 3.7 in the real and sham conditions, respectively) and the baseline condition (*p* < 0.05 at every second from the 31st to 36th second, with the highest difference observed at the 31st s, *p* = 0.01, corresponding to ∆OA values of 13.1 and 4.8 in the real and baseline conditions, respectively). Additionally, a significant difference in ∆OA values was observed at the 18th (*p* = 0.02) and 19th (*p* = 0.04) second between the baseline and sham sessions, indicating a less pronounced OA in the sham session. Conversely, there was no significant interaction observed for the ∆OA values recorded in the trigeminal region (Figure 6).

Regarding the forearm, considering that the ∆OA values during the real session differed significantly from the other two sessions starting from the final part of the T3 phase (compared to the sham session) or exclusively during the T4 period (compared to the baseline session), we decided to conduct a further targeted analysis. This aimed to evaluate the differences between sessions in early and late ∆OA values during T3, as well as after sensation during T4. For this purpose, we identified three distinct 6 s time windows: the 19th–24th second (T3A), the 25th–30th second (T3B), and the 31st–36th second (T4). We identified the 19th second as the start of the T3A period as this corresponded to the moment when the average ∆OA value stabilized in the baseline session in the forearm. Following this, a two-way ANOVA with within-subjects factors ‘condition’ (three levels: baseline, real, and sham) and ‘time’ (three levels: mean ∆OA values during the T3A, T3B, and T4 intervals) was conducted revealing a significant interaction of factors (F(4,84) = 4.2; *p* = 0.004) (Figure 7). Post-hoc analysis showed a significant difference in mean ∆OA values between the baseline and sham sessions during the T3A period (*p* = 0.018), and between the active session and both the baseline and sham sessions during both the T3B period (*p* = 0.048 and *p* = 0.002, respectively) and the T4 period (*p* = 0.00006 and *p* = 0.008, respectively).

Finally, ANOVAs performed to measure differences in the VAS values recorded during the CT and OA trials across the three different sessions for both the forearm (Figure 8) and the trigeminal region (Figure 9) showed a significant interaction between the factors in all cases, enabling post-hoc testing using Duncan’s test. The main results of the statistical analyses and the most significant differences in VAS values between CT and OA across different time periods are presented in Table 1. Notably, in the forearm, we observed that the post-active rTMS session demonstrated a significantly longer time interval (16 s) during which the VAS values during OA were significantly lower compared to CT, relative to both the baseline (9 s) and sham sessions (10 s).

## 4. Discussion

### 4.1. Key Findings

In the present study, we explored the modulation of the OA phenomenon through high frequency rTMS targeting the left M1 in a group of healthy subjects. We found that rTMS was well tolerated and did not significantly affect WDT, CDT, or Pain_50–60_. Conversely, a significant enhancement of the OA phenomenon was observed in the forearm after real rTMS during the late phase (T3B) and after-trial (T4) phases, compared to the baseline and sham sessions. This effect was not recorded in the trigeminal region, where no significant differences were observed across sessions.

### 4.2. Interpretation and Potential Action Mechanisms of rTMS

This study builds upon our previous work, which investigated the modulatory effects of a single high-frequency rTMS session targeting M1 on the OA measured in the thenar eminence [11]. In this latter study, we observed that the OA phenomenon was not detectable at baseline, but a delayed OA-like effect could be observed after active stimulation. The reason why the OA phenomenon could not be recorded at the thenar eminence is not clear, and two main hypotheses have been proposed: (1) the epidermal stratum corneum of the palm’s glabrous skin is at least twice as thick as that of non-glabrous skin, forming a more substantial barrier between nociceptors and the skin surface [39,40] and (2) variations in nociceptor innervation between the thenar eminence and non-glabrous skin [41,42]. Beyond the different anatomical and physiological characteristics of the stimulation site, it is important to note that in the present study we used a considerably larger total number of stimuli (3000 vs. 1200 stimuli) applied in longer trains and shorter interstimulus intervals, in accordance with the study by Attal et al. [11], hypothesizing that this would enhance the effectiveness of rTMS in modulating OA.

Though there is evidence that M1 is involved in pain modulation [11,43], the mechanisms through which M1 exerts its effects are still under discussion. It is hypothesized that stimulating the local motor cortex may alter the activity of brain regions located far from the stimulation site, especially areas related to the modulation of pain [44,45], thus inducing analgesic effects [35,46]. In particular, high-frequency rTMS over the M1 could induce long-term plasticity (LTP)-like responses, thus enhancing communication between M1 and pain modulatory centres like the anterior cingulate cortex (ACC) and the periaqueductal gray (PAG), strengthening descending inhibitory controls [13]. Furthermore, rTMS may modulate the inhibitory/excitatory balance to reduce hyperexcitability in pain pathways, with evidence suggesting the activation of GABAergic neurons projecting to the thalamus and modulation of pain-related thalamic activity [19,44]. Finally, recent studies investing functional connectivity also suggest that rTMS improves functional connectivity between M1 and key regions involved in pain modulation, such as the ventromedial prefrontal cortex, amygdala, posterior insula, and brainstem, potentially facilitating endogenous analgesic mechanisms [47]. These findings highlight the complex interplay between cortical and subcortical networks in rTMS-induced pain modulation.

The observed potentiation of OA in the late and after-trial phases suggests that the mechanisms activated by rTMS may be more responsive to prolonged painful stimuli. Additionally, the absence of early OA potentiation may be due to a ceiling effect where OA is maximally expressed under normal conditions and/or a nocebo effect, capable of masking the enhancement induced by stimulation. Indeed, the significant reduction in the magnitude of early OA observed after the sham session compared to baseline supports the hypothesis that a nocebo effect could develop in at least some subjects, even though they were neither positively nor negatively influenced by the examiner. In this regard, there is clear evidence that pain perception can be influenced by psychological and suggestion factors [48,49,50], and it has been observed that the OA phenomenon may be susceptible to hyperalgesic but not hypoalgesic suggestions [51]. On this basis, it is reasonable to assume that at least some of the subjects might have believed rTMS could have produced a hyperalgesic effect, resulting in the observed nocebo effect. These considerations strengthen the notion that the effect of real rTMS is genuine and linked to the activation of endogenous pain control mechanisms rather than placebo effects. In this regard, it is also worth noting that we stimulated the primary motor cortex (M1) instead of the DLPFC, which is more likely to produce placebo analgesia [52,53]. Furthermore, although we cannot completely rule out the possibility that the M1 stimulation might have activated neural circuits associated with a placebo response, despite no expectations being provided, it is noteworthy that Szikszay et al. [51] showed that positive expectations cannot enhance OA.

The lack of significant changes in the trigeminal territory could indicate a somatotopic specificity in the effects of rTMS. This finding might be related not only to the representation of the stimulated area (hand) in M1 but also to the neuroanatomical structure and functional properties of the cortical trigeminal region which could influence the response to rTMS. This finding may seem to contrast with previous studies suggesting that the analgesic effects of non-invasive brain stimulation can be widespread, regardless of the stimulation site in M1 [11,35,43]. However, some considerations should be made in this regard. In particular, while there is no evidence that different mechanisms control the OA phenomenon in different areas of non-glabrous skin, it should be noted that (1) the structural and functional properties of the trigeminal region might play a role in modulating the response to rTMS; (2) it has been observed that the analgesic effects of rTMS increase with repeated sessions [11], so it is possible that repeated sessions could have modulated the OA phenomenon even at the trigeminal level; and (3) the stimulation was applied to healthy subjects in whom physiological antinociceptive mechanisms are already normally expressed, making their potentiation difficult.

Future studies are needed to clarify whether the lack of significant changes in the trigeminal region is solely due to the lack of direct stimulation of the cortical representation of the head or if it would persist even with direct stimulation, possibly due to other underlying factors.

### 4.3. Safety, Technical Issues, and Study Limitations

Although rTMS is considered a safe and non-invasive technique, some participants may experience mild adverse effects, such as transient headaches or discomfort at the stimulation site. However, in our study, none of the participants reported any adverse effects during or after the experimental sessions. Earplugs were provided to minimize auditory discomfort during stimulation, and the relatively short duration of each session (15 min) ensured that participants could maintain a stable head position without difficulty, supported by the use of a headrest. These measures likely contributed to the overall tolerability of the procedure. It is important to note that our study was conducted on young, healthy individuals undergoing single sessions of rTMS. We did not evaluate the safety and tolerability of repeated sessions, nor did we include older individuals or patients with chronic pain syndromes in our assessment. However, previous evidence [11] shows that rTMS is well-tolerated even in neuropathic pain patients, with only mild and transient side effects, and no serious adverse events, supporting the overall safety of the procedure.

Several considerations and limitations of this study should be addressed. Firstly, the hand motor hotspot was identified without the aid of neuronavigation, leaving open the possibility that more pronounced effects might have been achieved with active rTMS if image-guided navigation had been used to target the anterior border of the central sulcus. However, there is currently no evidence suggesting that neuronavigation leads to better outcomes in pain therapy with M1-rTMS [54,55]. Second, a sham coil was not used, so we cannot exclude the possibility that some subjects may have noticed differences between the real and sham sessions. Nevertheless, hypothesizing a shift from a nocebo effect (observed after sham rTMS) to a placebo effect (observed after real rTMS) seems difficult to propose, especially considering that no positive or negative expectations were given to the subjects. Third, the reliability of the OA paradigm was not assessed in this study. However, other researchers have provided evidence suggesting that it may be robust [51,56].

Additionally, the sample size of this study was relatively small, which may result in limited statistical power. Future studies should include larger sample sizes to improve the stability and generalizability of the findings. Furthermore, this study was conducted exclusively in healthy subjects, and its applicability and effectiveness in chronic pain patients remain to be validated in future research. These considerations are critical to ensuring that the observed effects can be translated to clinical populations.

Lastly, technical limitations and individual variability must also be considered. The lack of neuronavigation and the inherent variability in cortical anatomy may have influenced the precision of stimulation, potentially contributing to variability in individual responses. While the motor hotspot method is widely used, it may not account for subtle differences in cortical organization among individuals, which could affect the reproducibility of effects across subjects. Addressing these technical and methodological limitations in future studies will help refine rTMS protocols and enhance their efficacy and reliability.

### 4.4. Clinical Implications and Suggestions for Future Research

Based on our findings, high-frequency rTMS targeting M1 for pain modulation should, whenever possible, be applied to the cortical representation area of the affected body region rather than universally to the hand representation of M1. If confirmed by future studies applying rTMS to cortical areas other than the hand representation area and demonstrating therapeutic effects or enhanced OA, this approach could maximize therapeutic benefits while minimizing the risk of the treatment being ineffective.

Ultimately, the current findings in healthy individuals, where the mechanisms and anatomical pathways underlying the OA phenomenon operate normally, cannot be directly applied to patients with different chronic pain conditions, where OA can be compromised [32,35]. Therefore, future studies are needed to evaluate the modulatory effects of rTMS on OA in these patient populations (e.g., patients with migraine or fibromyalgia) to better understand the therapeutic potential and mechanisms of action of rTMS. Future research should also explore the use of functional magnetic resonance imaging (fMRI) and other techniques, such as near-infrared spectroscopy (NIRS) and TMS–electroencephalography (TMS-EEG), to monitor changes in brain activity during and after rTMS stimulation. These approaches could provide complementary insights into the neural correlates of rTMS and enhance its application. Investigating the effects of stimulating other areas, like the DLPFC, is also crucial, although Attal et al. [11] found M1 to be more effective for treating chronic pain. Additionally, it is important to assess the role of psychological and cognitive factors in shaping responses to rTMS. Identifying predictive factors for treatment response could help tailor interventions more effectively. For example, evaluating whether different cortical targets might be optimal for patients with specific psychological or cognitive characteristics could refine therapeutic strategies. Furthermore, further studies are needed to understand how demographic factors, including gender and age, impact rTMS responses.

### 4.5. Significance and Conclusions

In conclusion, this study adds to the growing body of evidence suggesting that rTMS can modulate central pain mechanisms, specifically the OA phenomenon, in healthy subjects, and this modulation appears to be region-specific. The findings underscore the potential of rTMS applied to M1 not only as a therapeutic intervention for chronic pain but also as a means to deepen our understanding of the complex mechanisms underlying the brain’s regulation of pain. These insights pave the way for improving the efficacy of innovative pain management strategies based on rTMS, with potential for broader applications within the field of pain treatment, including optimizing neuroplasticity, refining motor rehabilitation approaches, and addressing sensory dysfunctions associated with pain conditions.

## Figures and Tables

**Figure 1 life-15-00182-f001:**
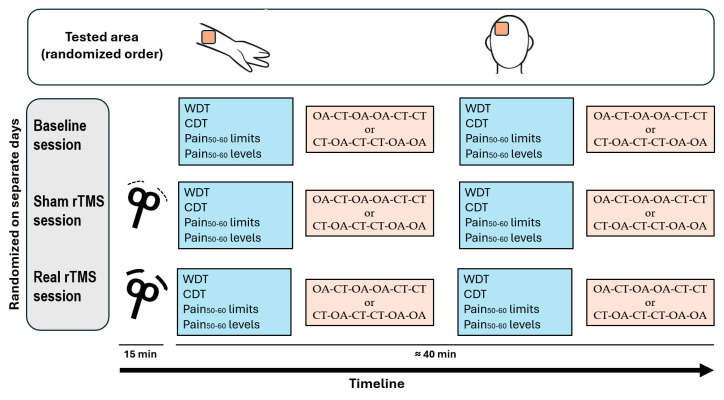
The flowchart outlines the study design to investigate the effects of repetitive transcranial magnetic stimulation (rTMS) on the offset analgesia (OA) phenomenon. This study consists of three main sessions (baseline, sham, and real rTMS session), that were randomized and conducted on separate days. During each session, first, sensory and pain thresholds were assessed, followed by a sequence alternating between offset analgesia (OA) and constant temperature (CT) trials. The tested areas, including the forearm and trigeminal regions, were assessed in a randomized order to evaluate whether rTMS effects are region-specific. Cold detection threshold: CDT; Pain_50–60_: heat pain threshold corresponding to a VAS score of approximately 50–60 out of 100; warm detection threshold: WDT.

**Figure 2 life-15-00182-f002:**
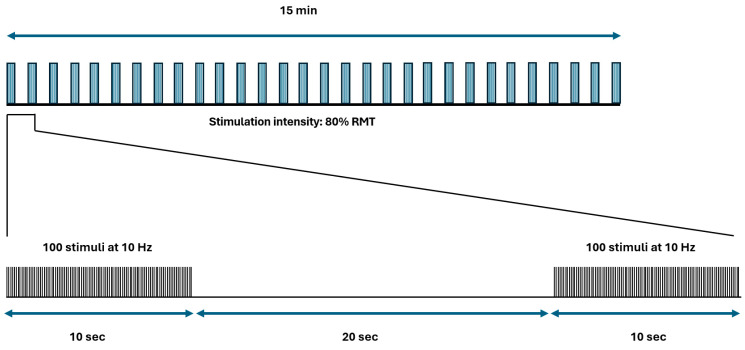
The diagram represents the repetitive transcranial magnetic stimulation (rTMS) protocol utilized in this study, detailing the sequence and pattern of stimulus application. RMT: resting motor threshold.

**Figure 3 life-15-00182-f003:**
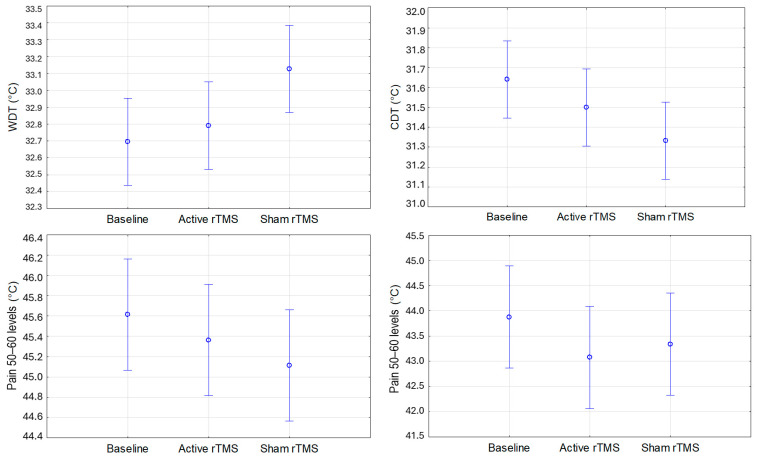
Mean values for WDT, CDT, and heat pain threshold corresponding to Pain_50–60_ assessed by the methods of levels and limits at baseline and after active (real) and sham (placebo) rTMS in the forearm. Vertical bars denote 95% confidence intervals. WDT: warm detection threshold. CDT: cold detection threshold. rTMS: repetitive transcranial magnetic stimulation.

**Figure 4 life-15-00182-f004:**
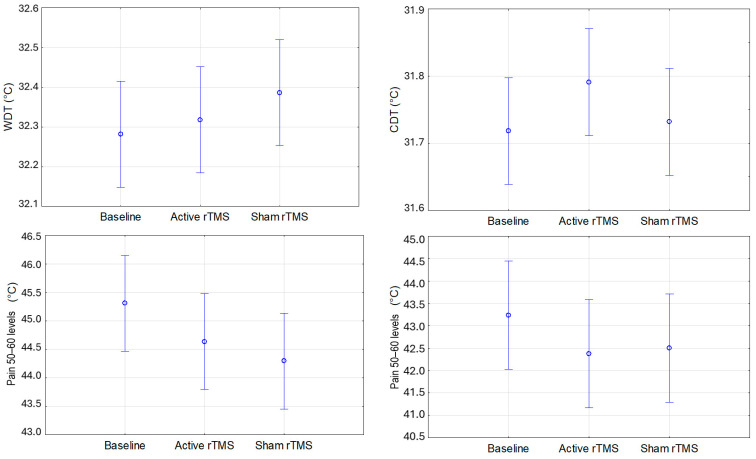
Mean values for WDT, CDT, and heat pain threshold corresponding to Pain_50–60_ assessed by the methods of levels and limits at baseline and after active (real) and sham (placebo) rTMS in the forehead. Vertical bars represent the 95% confidence intervals. WDT: warm detection threshold. CDT: cold detection threshold. rTMS: repetitive transcranial magnetic stimulation.

**Figure 5 life-15-00182-f005:**
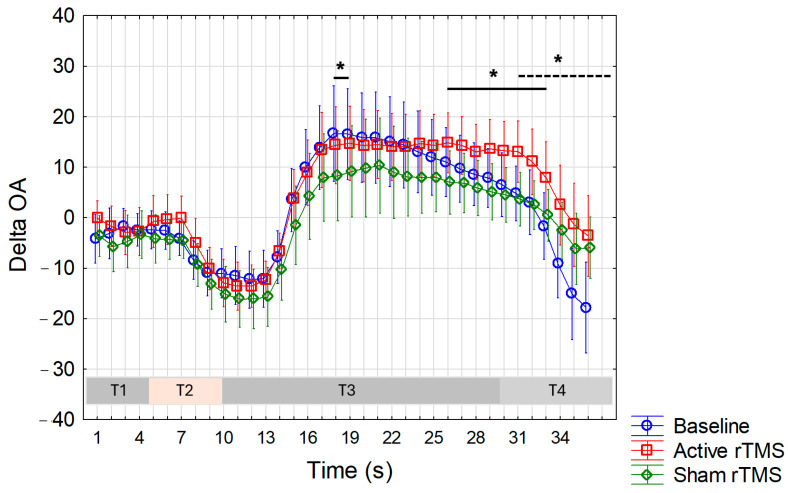
The magnitude of the OA phenomenon in the forearm was determined by subtracting the VAS values recorded at 1 s intervals during the OA trial from those obtained during the constant trial (∆OA) at baseline, and after active (real) and sham (placebo) rTMS. Note that during the T3 period, positive values indicate an OA phenomenon that is more pronounced the higher the value. Mean values are reported. Vertical bars represent the 95% confidence intervals. T1, T2, T3, and T4 correspond to distinct time intervals within the OA trial (see details within the text). OA: offset analgesia; s: seconds. The long solid line indicates significant differences between ∆OA values in the active condition versus the sham condition; the short solid line indicates significant differences between ∆OA values in the baseline condition versus the sham condition; the dashed line indicates a significant difference between ∆OA values in the active condition versus the basal condition. * Indicates significance at *p* < 0.05.

**Figure 6 life-15-00182-f006:**
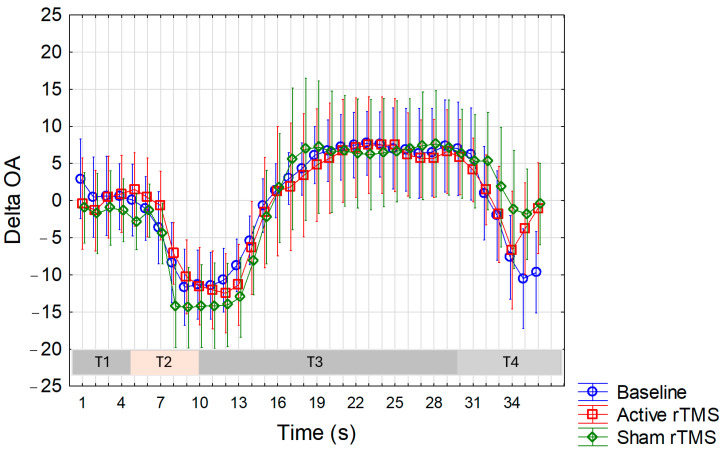
The magnitude of the OA phenomenon in the forearm was determined by subtracting the VAS values recorded at 1 s intervals during the OA trial from those obtained during the constant trial (∆OA) at baseline, and after active (real) and sham (placebo) rTMS. Note that during the T3 period, positive values indicate an OA phenomenon that is more pronounced the higher the value. Mean values are reported. Vertical bars represent the 95% confidence intervals. T1, T2, T3, and T4 correspond to distinct time intervals within the OA trial (see details within the text).

**Figure 7 life-15-00182-f007:**
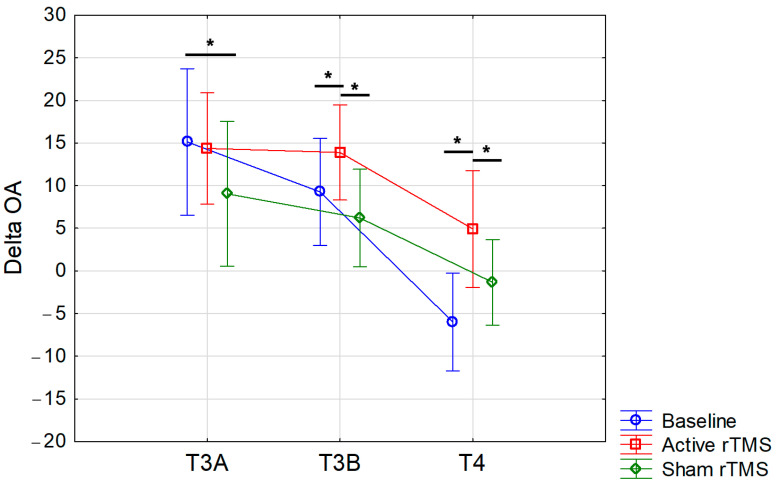
Mean delta OA (∆OA) values were calculated for 6 s time intervals during the T3 and T4 periods at baseline and after active (real) or sham (placebo) rTMS. Positive values indicate an OA phenomenon that becomes more pronounced as the value increases. Vertical bars denote 95% confidence intervals. T3A, T3B, and T4 correspond to the following three distinct 6 s time windows: 19th–24th second (T3A), 25th–30th second (T3B), and 31st–36th second (T4). The solid lines indicate significant differences. * Indicates significance at *p* < 0.05.

**Figure 8 life-15-00182-f008:**
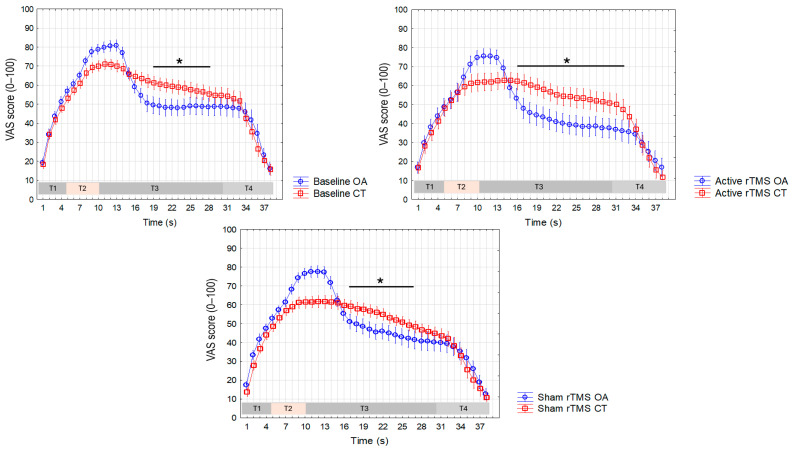
Visual analogue scale (VAS) values recorded during the constant trial (CT) and the offset analgesia trial (OA) in the forearm across the three experimental sessions (baseline, active rTMS, and sham rTMS). Mean values are presented, with vertical bars representing the standard error of the mean. The horizontal bar with an asterisk indicates significant differences (*p* < 0.05 at each 1 s time point).

**Figure 9 life-15-00182-f009:**
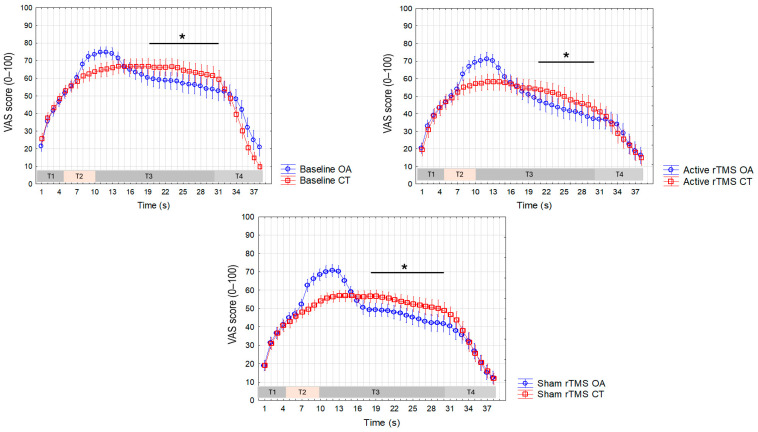
Visual analogue scale (VAS) values recorded during the constant trial (CT) and the offset analgesia trial (OA) in the trigeminal region across the three experimental sessions (baseline, active rTMS, and sham rTMS). Mean values are presented, with vertical bars representing the standard error of the mean. The horizontal bar with an asterisk indicates significant differences (*p* < 0.05 at each 1 s time point).

**Table 1 life-15-00182-t001:** Statistical results of the two-way ANOVA interaction and significant differences in visual analogue scale (VAS) values between the constant trial (CT) and offset analgesia (OA) trials across experimental sessions (baseline, active rTMS, and sham rTMS) for both the forearm and trigeminal region. For each session and body region, the table specifies the time intervals showing significant differences (highlighted in bold) and the corresponding maximum differences observed during the T3A, T3B, and T4 phases. All reported *p*-values indicate the level of statistical significance.

Body Region	Session	Interaction Between Factors	Significant VAS Values, Differences Between CT and OA Trials	Max T3A Phase Difference	Max T3B Phase Difference	Max T4 Phase Difference
**Forearm**	**Baseline**	F(3,66) = 9.2, *p* < **0.00001**	From 19th s to 28th s	At 19th s (*p* = **0.00004**)	At 25th s (*p* = **0.004**)	At 31st s (*p* = 0.07)
	**Active**	F(4,80) = 16.4, *p* < **0.00001**	From 16th s to 32nd s	At 24th s (*p* = **0.000001**)	At 26th s (*p* = **0.000001**)	At 31st s (*p* = **0.0002**)
	**Sham**	F(3,67) = 11.8, *p* < **0.00001**	From 17th s to 27th s	At 21st s (*p* = **0.0002**)	At 25th s (*p* = **0.006**)	At 31st s (*p* = 0.2)
**Trigeminal Region**	**Baseline**	F(3,60) = 11.1, *p* < **0.00001**	From 19th s to 31st s	At 23rd s (*p* = **0.001**)	At 29th s (*p* = **0.001**)	At 31st s (*p* = **0.007**)
	**Active**	F(3,55) = 7.3, *p* < **0.00001**	From 21st s to 30th s	At 24th s (*p* = **0.008**)	At 25th s (*p* = **0.008**)	At 31st s (*p* = 0.15)
	**Sham**	F(2,49) = 9.7, *p* < **0.00001**	From 18th s to 30th s	At 19th s (*p* = **0.01**)	At 28th s (*p* = **0.004**)	At 31st s (*p* = 0.05)

## Data Availability

Raw data used in this study will be available in the Zenodo repository at https://doi.org/10.5281/zenodo.14288005.

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
