# Peer review of "Effects of Repetitive Transcranial Magnetic Stimulation Applied over the Primary Motor Cortex on the Offset Analgesia Phenomenon"

_life, 2025, doi:10.3390/life15020182_

Round 1
Reviewer 1 Report
Comments and Suggestions for Authors
This study examined the effects of high-frequency repetitive Transcranial Magnetic Stimulation (rTMS) over the left upper limb primary motor cortex (M1) on the Offset Analgesia (OA) phenomenon, a measure of endogenous pain modulation, in 22 healthy volunteers across baseline, active, and sham sessions. It aimed to determine whether rTMS effects on OA were specific to the forearm, whose cortical area was stimulated, or generalized to other regions like the trigeminal area. Key findings revealed that active rTMS enhanced OA in the forearm during the late phase, with no significant effects in the trigeminal region, suggesting region-specific modulation of central pain mechanisms linked to M1's somatotopic organization.
While this study offers valuable insights into rTMS applications for chronic pain management, several critical improvements are required to enhance clarity, coherence, and depth. Below are the detailed critiques and my recommendations for improvement.
Major Critiques and Suggestions:
1. The study lacks a clearly defined hypothesis or explicit objectives in its abstract and introduction. This omission reduces engagement and makes it challenging for readers to understand the study’s focus. Clearly state the hypothesis and objectives early in the manuscript, linking them to the rationale for examining region-specific effects of rTMS on OA.
2. The methodology, particularly for QST assessments and stimulation procedures, is not presented in a reader-friendly manner. This hampers reproducibility and reader comprehension.
- Include a flowchart detailing the steps of QST assessments.
- Provide a separate flowchart or diagram outlining the rTMS stimulation procedures, including parameters such as frequency, intensity, and duration.
3. The interpretation of the result is dense and difficult to follow, which diminishes the impact of the findings.
- Simplify the discussion by breaking it into subsections with clear headings (e.g., "Key Findings," "Interpretation," "Clinical Implications").
4. The manuscript lacks a detailed discussion of the potential mechanisms by which rTMS enhances OA.
- Elaborate on how rTMS might modulate cortical or subcortical pain pathways, potentially involving synaptic plasticity, changes in inhibitory/excitatory balance, or altered functional connectivity.
5. The manuscript does not address the possible off-target effects of rTMS or limitations related to adverse effects and technical challenges.
- Acknowledge and discuss the potential adverse effects, such as discomfort or fatigue, and their implications for clinical use.
- Address technical limitations, including the precision of stimulation and variability in individual responses.
Comments on the Quality of English LanguageThe English could be improved to express the research clearly.
Author Response
Dear Reviewer,
the point-by-point responses to your comments can be found in the attached pdf file

Reviewer 2 Report
Comments and Suggestions for Authors
The following is a summary of this academic paper
1. The purpose is not clearly stated, the methodology is too briefly described, the results are not specifically stated, and the conclusions are too generalized.
2. When presenting the complexity of pain, in addition to neural activity, the relationship between pain and psychological factors (e.g., anxiety, depression, etc.) should be added, as well as the mechanisms of how these psychological factors may exacerbate pain by influencing brain function and thereby giving the reader a more comprehensive understanding of the multidimensional nature of pain .
3. When discussing endogenous pain modulation mechanisms, in addition to mentioning the phenomena of Conditioned Pain Modulation (CPM) and Offset Analgesia (OA), more recent studies should be cited to explore the specific manifestations and differences in the roles of these mechanisms in different types of pain (e.g., neuropathic pain, inflammatory pain, etc.) to provide a more theoretically rich basis for the subsequent studies. Provide a richer theoretical foundation for subsequent studies.
4. It is suggested to add a paragraph at the end of the introduction to clearly point out the gaps in the current research on the effects of rTMS on OA phenomenon, such as “it is still unclear whether there is a difference in the modulation effect of rTMS on OA phenomenon in different body regions and what are the underlying neurological mechanisms”, so as to provide a stronger support for the necessity and innovativeness of the present study. The hypotheses of the study are not clearly stated.
5. It is recommended that the hypothesis of this study be clearly stated at the end of the introduction, such as “We hypothesize that high-frequency rTMS can enhance the OA phenomenon in the forearm region, but has less effect on the OA phenomenon in the trigeminal region”, so that readers will have a clearer understanding of the direction of the study and the expected results.
6. Insufficiently detailed description of the screening criteria: In the exclusion criteria, it is suggested to add “recent use of drugs that may affect neuroexcitability, such as antiepileptic drugs, antidepressant drugs, etc.” to ensure that the baseline state of the subjects is more stable and to minimize the interference of drug factors on the experimental results.
7. The description of the experimental process is not clear enough. When describing the experimental process, it is recommended to use a flow chart or timeline to visually display the specific steps and time schedule from the time the subject enters the laboratory to the completion of all the tests, such as “The subject first performs the QST test (about 30 minutes), then receives rTMS stimulation (15 minutes), and finally performs the QST test again (30 minutes) ”, making it easier for the reader to understand the organization of the experiment.
8. When describing the QST assessment, it is recommended to add specific procedures and precautions for each test, such as “During the measurement of thermal pain thresholds, the temperature rises at a rate of 1°C/s, subjects need to press the button as soon as they feel the pain reaching a VAS score of 50-60, and they need to keep their arm still during the test”, to ensure that other researchers were able to accurately replicate the experimental conditions of this study.
9. The description of the rTMS stimulation parameters was not comprehensive enough. In addition to the stimulation frequency, number of pulses, and intensity, information such as the type of stimulation coil and cooling method should be added, such as “70 mm figure-of-eight water-cooled coils equipped with the Magstim Rapid2 Stimulator were used”, as well as the subject's position and head immobilization during stimulation, to ensure the reproducibility of the experiments. Reproducibility.
10. The method of data processing was not described in sufficient detail. When describing the statistical analysis, it is suggested to add the specific steps of data pre-processing, such as “firstly, all the data were tested for normality using the Kolmogorov-Smirnov test, and then data that did not conform to the normal distribution were subjected to a logarithmic transformation or a non-parametric test” to ensure the rigor of the data analysis. rigor of data analysis .
11. Data presentation is not intuitive enough. It is suggested to add more charts in the results, such as bar charts, line graphs, etc., to visualize the changes of each index under different experimental conditions and the results of intergroup comparison, so that readers can understand and compare the data more easily.
12. The description of the results is not specific enough. When describing the effect of rTMS on the OA phenomenon, in addition to mentioning the significant difference, it should also specify at which time points or time periods the difference is most obvious, e.g., “At the 28th second of the T3B stage, the average ∆OA value of the real rTMS group was X points higher than that of the sham rTMS group, reaching statistical significance (p<0.05) “ to provide more detailed timing information .
13. Interpretation of results was not in-depth enough. In the results section, it is recommended that some unexpected or inconsistent results be explored in more depth, such as “No significant difference was observed in the trigeminal region, which may be related to the neuroanatomical structure and functional properties of this region, and future studies may further explore the reasons for this”, to guide the reader to think about the possible mechanisms and influencing factors.
14. The discussion of the significance of the results is not comprehensive enough. When discussing the significance of rTMS on the enhancement of OA phenomenon, in addition to the treatment of chronic pain, its potential applications in other fields should also be considered, such as “This finding provides new ideas for the development of pain management strategies based on rTMS, and may be important for understanding the complex mechanisms of the brain's role in the regulation of pain”, to expand the breadth and depth of the discussion. The breadth and depth of the discussion.
15. Mechanisms are not explored in sufficient depth. It is suggested to combine the latest neuroscience research to further explore the specific neural circuits and neurotransmitter systems that may be activated by rTMS, such as “rTMS may activate glutamatergic neurons in the M1 region, which may affect the inhibitory effect of γ-aminobutyric acid (GABA) neurons in the downstream pain pathway”, providing a clearer direction for the subsequent research. The study limitations were not sufficiently analyzed.
16. Limitations of the study were not sufficiently analyzed. In addition to the limitations mentioned above, we should also add “the sample size of this study is relatively small, which may have certain problems of insufficient statistical efficacy, and future studies should expand the sample size in order to improve the stability and generalizability of the results” and “the study was conducted only in healthy subjects, and its applicability and effect in chronic pain patients need to be further verified”. The study was only conducted in healthy subjects, and its applicability and effectiveness in chronic pain patients need to be further validated”, which gives the reader a more comprehensive understanding of the limitations of the study.
17. Suggestions for future research directions were not specific enough. It is suggested that more specific suggestions for future research should be made, such as “future research can use functional magnetic resonance imaging (fMRI) and other neuroimaging techniques to monitor the changes in brain activity during rTMS stimulation in real time, so as to reveal more intuitively its influence on the pain regulation mechanism”, so as to provide a clear idea and methodology for the subsequent research.
Author Response

(The authors gave the same response as above.)

Round 2
Reviewer 1 Report
Comments and Suggestions for Authors
I am pleased to acknowledge the author's thorough consideration of the reviewer’s suggestions, as demonstrated in the revised manuscript. The revisions effectively address all the raised concerns and incorporate the recommended changes, resulting in notable improvements to the manuscript's clarity and organization. These enhancements, particularly in presenting complex technical discussions, greatly improve its accessibility and readability. This dedicated effort significantly elevates the overall quality of the article.